# Possibilities of Increasing the Durability of Dies Used in the Extrusion Process of Valve Forgings from Chrome-Nickel Steel by Using Alternative Materials from Hot-Work Tool Steels

**DOI:** 10.3390/ma17020346

**Published:** 2024-01-10

**Authors:** Marek Hawryluk, Marta Janik, Maciej Zwierzchowski, Marzena Małgorzata Lachowicz, Jakub Krawczyk

**Affiliations:** 1Department of Metal Forming, Welding and Metrology, Wroclaw University of Science and Technology, Lukasiewicza 5 Street, 50-370 Wroclaw, Poland; marta.janik@mahle.com (M.J.); maciej.zwierzchowski@pwr.edu.pl (M.Z.); marzena.lachowicz@pwr.edu.pl (M.M.L.); jakub.krawczyk@pwr.edu.pl (J.K.); 2MAHLE Polska, Mahle 6, 63-700 Krotoszyn, Poland

**Keywords:** durability of forging tools, extrusion process, NCF 3015 steel, hot-work tool steels

## Abstract

This study refers to an analysis of the dies used in the first operation of producing a valve forging from chromium-nickel steel NC3015. The analyzed process of manufacturing forgings of exhaust valves is realized in the co-extrusion technology, followed by forging in closed dies. This type of technology is difficult to master, mainly due to the increased adhesion of the charge material to the tool substrate as well as the complex conditions of the tools’ operations, which are caused by the cyclic thermo-mechanical loads and also the hard tribological conditions. The average durability of tools made from tool steel WLV (1.2365), subjected to thermal treatment and nitriding, equals about 1000 forgings. In order to perform an in-depth analysis, a complex analysis of the presently realized technology was conducted in combination with multi-variant numerical simulations. The obtained results showed numerous cracks on the tools, especially in the cross-section reduction area, as well as sticking of the forging material, which, with insufficient control of the tribological conditions, can cause premature wear of the dies. In order to increase the durability of forging dies, alternative materials made of hot work tool steels were used: QRO90 Supreme, W360, and Unimax. The preliminary tests showed that the best results were obtained for QRO90, as the average durability for the tools made of this steel equaled about 1200 forgings (with an increase in both the minimal and maximal values), with reference to the 1000 forgings for the material applied so far.

## 1. Introduction

As the automotive market has been aiming to meet the customers’ demands, there is a search for new materials that would exhibit excellent high-temperature creep resistance and heat resistance as well as gas corrosion and oxidation resistance [1]. This is dictated by the restricted norms of combustion gas emissions and the introduction of a lean fuel mixture into the combustion technology, which has stimulated the development of new materials for car engine valves [2,3]. Currently, such materials fulfilling the requirements are chromium-nickel steels, e.g., steel NC3015, with the commercial name Nireva, which, due to the high price of nickel, is the cheapest equivalent of nickel-based super alloys [4]. This steel is characterized by an austenitic structure with a γ’ and γ” phase (Ni3Nb), with the main phase reinforcing the alloy [5]. Nireva contains an increased content of Cr and Al, which play the dominant function in increasing the corrosion resistance, and additionally, the aluminum improves the alloy’s resistance to hot oxidation and is also part of the composition of the intermetallic phase Ni3(Al,Nb), increasing the material’s heat resistance [6]. Due to a higher affinity of C to Ti and Nb, with which it forms large amounts of carbides in the alloy, chromium can react with oxygen from the atmosphere on the alloy’s surface and form a layer of chromium oxides [6,7]. Unfortunately, the presence of intermetallic phases in steel used for valves, which increases the resistance to high temperatures, significantly hinders the processes of forming this austenitic steel in plastic forming processes, with respect to forging carbon steels, worsening its deformability, and additionally, the present hard particles lead to more rapid tool wear [8,9,10]. Currently, two main valve manufacturing technologies are applied in plastic forming processes. One of them is forging in closed dies, which makes it possible to obtain better forging with respect to quality and performance, yet it is insufficiently mastered with regard to the other, much better-mastered technology, consisting in the electro-upsetting of a small-diameter charge material. However, the latter is more expensive, and the obtained products are characterized by worse properties. That said, we should mention that there are other methods of producing engine valves, especially hollow drilled valves, which are constructed in a way that makes them light and allows them to conduct heat along the valve [11], or valves produced through rolling and wedge forging [12], and although they are well-mastered, due to the worse properties of the forgings and high costs of the particular technologies, they are unprofitable. For this reason, in the case of the technology of manufacturing valve forgings through hot forging in closed dies, the key is the selection of the proper technological parameters, e.g., heating of the charge material [13], tribological conditions, including lubrication [14], optimal tool construction ensuring reduced forming forces, as well as minimized residual deformations. That is because an insufficiently mastered technology is the main cause of shorter lifespans of forging tools and instrumentation. Sizable problems include high cyclic pressures (of the order of over 2500 MPa), intensive friction, and difficult tribological conditions, especially in the aspect of the forming tools. In effect of such conditions as well as the result of a long path of friction, we often observe the blocking or jamming of the extruded material in the eye of the die [15]. The problem of durability of extruding and forging tools results also from the increased adhesion of the charge material made from chromium-nickel steel (Ni 25–35% and Cr 15–20%) to the substrate of a tool made of tool steel [16]. All this causes premature wear and reduced tool durability. The relatively short life of tools used to produce valves is a consequence of the occurrence of many phenomena and wear mechanisms taking place at varied frequencies. Additionally, the cyclic thermo-mechanical loads exerted on the forging tools, as well as the dynamics and instability of the forging process, make it a production process that is one of the most difficult to analyze. For this reason, it is justified to conduct advanced complex studies making it possible to analyze the destructive phenomena and mechanisms. On this basis, it is possible to undertake corrective as well as preventive measures and propose methods enabling an increased durability of forging tools. This makes it possible to reduce the unit costs of producing valve forgings and other elements manufactured in die-forging processes [17]. The development of methods increasing the durability of forging tools is a multi-variant one and brings measurable effects. Technologists and engineers have been working continuously as a whole to improve the tools as well as their surface layers and working environments. In the case of the tools themselves, it is crucial to properly select the material for hot operations and its thermal treatment or, more rarely used, sub-zero treatment. It is as important to properly prepare the construction and shape of the die, which also affects the life of the forging tools. Another group of methods improving the performance parameters of the tools is the modification of the surface layer, among which the most frequently applied is thermo-chemical treatment, especially nitriding. Welding, bundle, and hybrid techniques as well as mechanical methods are also known and popular. The last group of methods increasing tool life is connected with the working environment, that is, the cooling, lubrication, as well as automatization and robotization of the process [18].

Currently, a number of IT engineering tools and numerical methods based on FEM [19,20,21] are used to analyze and optimize industrial forging processes, often combined with thermal imaging measurements as well as microstructural research or dimensional analysis using laser scanners [22,23]. However, it seems that the most information can be obtained from numerical modeling, because such computational packages allow the determination of many physical quantities and other technological parameters that are difficult or even impossible to determine experimentally [24,25,26], and by introducing new functions verified in industrial conditions, they allow for quick the analysis of the entire industrial process, e.g., determining the distribution of temperatures, stresses, forging forces, flow errors of the deformed material, and even defects in forgings and many other technological aspects [27,28]. Currently, the combination of the above-mentioned research and measurement methods, in particular using increasingly advanced simulation programs based on FEM and FVM, enables comprehensive analysis on the basis of which it is possible to solve many scientific and technological issues [29,30,31].

One of the superior parameters selected during the construction of tools used for hot extrusion and forging is the selection of steel for hot operations, as it is one of the cheapest and most relatively easy-to-analyze methods of improving forging tools’ durability [32]. The conditions of forging tool operations require technologists to properly select the tool material so that it can maximally fulfill the expectations related to their work [33,34]. Among the available hot operation tool steels, no specific grade can simultaneously meet all these demands, one of the reasons being that some of them are contradictory [35,36]. So, in order to properly choose the tool material, we need to largely depend on the experience of the tool steel producers as well as the users of the tools, i.e., die forges [37,38]. At the same time, we should emphasize that, among the whole spectrum of available steels, there is none that would fulfill the expectations of the production in a complex way; so, while choosing the tool material [39], we can mostly base our decisions on the experience of the steel producers as well as the people directly connected with the production process [40]. The most common forging tool material used in forging processes are warm and hot operation tool steels, that is, 1.2343, 1.2344, 1.2367, 1.2999, etc., which are characterized by very good mechanical properties (high tensile strength and hardness, high abrasion resistance, high yield point of 2200 MPa). A typical thermal treatment for these steel grades consists of hardening and two- or three-fold tempering [41,42]. In these steels, the secondary hardness effect is used, which occurs at a tempering temperature of about 500 °C. At present, specialized tool steels are also introduced (e.g., alvar, dievar, hotvare, orvar suppreme, unimax, viadar, and thermodur 2367 or 2999) as well as other alternative tool materials [43,44,45]. At the same time, it should be emphasized that there is no material that would be “ideal” for most industrial die-forging processes, because, as suggested by the studies, the use of the same material for different tools in different processes can bring entirely different results. Also, every production process should be approached individually as the current operation conditions of the tool may either require a combination of the known tool durability-increasing techniques or can cause the necessity of developing new methods. For this reason, it is fully justifiable to perform further continuous research in the areas of selecting the optimal tool material for a specific forging process and even for the particular forging operation, especially in the case of producing forgings from austenitic steels or nickel alloys [46].

## 2. Materials and Methods

The subject of this research is a detailed analysis of the durability of the forging tools used in the first operation of the two-operation process of producing valve forgings together with the proposal of a method for increasing tool life by way of introducing other tool materials. The analyzed process involves the use of two pairs of tools: dies and punches, and for each of them, their wear was determined based on the real production process. In the first operation, the blocking die is the one that becomes worn first (Figure 1b), whereas the punch hardly wears at all (Figure 1a). In the second operation, the case is reversed, as the forging’s stem is already formed in the first operation, and the die in the second operation wears much slower (Figure 1d) than the central protrusion on the punch—the calotte (Figure 1c). So, a detailed analysis was performed on the forging dies applied in operation I of the hot co-extrusion process.

The blocking dies in the analyzed production process are made of the WLV material. These tools are subjected to standard thermal treatment, i.e., hardening and two-fold tempering to the hardness of 52–53 HRC; next, they undergo gas nitriding at a temperature of about 510 °C for 5 h, to reach a diffusion layer thickness of about 0.15 mm. The tool is mounted in a casing heated to about 200 °C and is next placed in the seat of the press (Figure 2). During the extrusion process, the blocking die is lubricated with oil-based graphite by means of a ring lubricator, mounted on the extrusion die’s casing. The temperature of the charge material equals 1040 °C. The tool prepared in this way is mounted on the press bench. The plastic forming process is realized on an eccentric press with a maximal pressure of 700 tons, whereas, because two operations are simultaneously conducted in one press movement, the total pressure applied in both operations equals over 300 tons.

The transport of the heated material inside the press takes place by means of a “basket” to the first seat, where the bearing roller is dropped into the blocking die, in which the valve’s stem is formed with the use of a hot extrusion process. Next, by means of the manipulator’s grippers, the ready slug forging is transported to the die located in the second seat, where the forging process takes place and the disk and profile of the valve are formed, whose shape is close to that of a ready product. In the industrial process, lubrication in the first operation for the analyzed tools (dies) is carried out by a special ring (flat sleeve) mounted in the die housing. The lubricant is a mixture of graphite and oil in the proportions of 1:12 and is fed automatically through a special lubrication device. Due to the lowest durability of all four tools in the set, a detailed analysis was performed on the dies used in operation I, for which the average operation time equals about 1000 forgings. In order to investigation and analysis, the following research techniques/methods were used:−macroscopic tests combined with a measurement of the wear/allowance on the tools’ working surface through 3D scanning by means of a laser scanner with the measuring arm ROMER Absolute ARM 7520si (Stockholm, Sweden) integrated with a RS3 and a comparison of the scan’s geometry with the CAD models;−FEM (Finite Element Method) modeling of the working conditions and temperature performed in the Forge 3.0 NxT program;−observations of the changes occurring on the working surface by means of a scanning electron microscope (SEM) with magnifications over 1000× by using LEO 1430 coupled with an EDX detector (Zeiss, Jena, Germany);−microstructural tests conducted in the surface layer of the tool’s cross-section by the light microscopy method after its etching in natal by means of a stereoscopic microscope Keyence VHX-S600E (Osaka, Japan) and a light microscope Olympus BX51M (Tokyo, Japan);−microhardness tests on the cross-section as a function of the distance from the surface with the use of a LECO microhardness tester (LECO Corporation, St. Joseph, MO, USA).

## 3. Results and Discussion

The investigations were divided into several stages, with the first step being a complex analysis of the currently realized technology, followed by the use of numerical modeling to determine the key parameters and physical quantities for a more thorough analysis of the process. The final stage was the introduction of alternative tooling materials to increase durability.

### 3.1. Analysis of the Current Technology

The technology of forging in closed dies applied in the analyzed process, despite its obvious advantages referring to the quality and performance properties of the forging, is rarely used because it is a process that is very difficult to master and that requires the preparation of the charge material mass with the precision of ±1–2% and maintaining a constant charge heating temperature as well as a proper design of the geometry of the die working impressions, ensuring their proper filling. The biggest problems of the analyzed technology of forging in closed dies are, however, the very high pressures occurring in the forging process, which cause problems with the low durability of the die (in some cases, even up to a few tens of forgings and in extreme cases, up to one forging) and the creation of faulty products resulting from the difficulties in the forming of steel with a high nickel content as well as the selection of the optimal process parameters. One of the biggest difficulties is the increased adhesion of the charge material made of nickel-chromium steel (nickel content up to 25–35%, chromium content of about 15–20%) to the substrate of the tool made of tool steel WLV during the process of forward extrusion. Another important problem is the lack of dissolution of the carbides in the charge material in the case of improper charge heating (for this steel, the temperature should be within the scope of 1040–1080 °C; a temperature that is too low causes a lack of dissolution of the hard carbide fractions and a temperature that is too high causes burning of the material). The high pressures and temperatures as well as the long friction path cause blocking of the extruded material in operation I in the eye of the die. Based on the data collected from the real production process, the average minimal and maximal durability for the blocking die was determined (Figure 3).

The diagram presents the durability data for 46 tools that were consecutively removed from the production process. The minimal value is at the level of 1 forging, whereas the maximal value equals 2436 items. The average wear is at the level of 1189 forgings. The tools for which the minimal and maximal values were obtained became the subject of analysis and contributed to the selection and evaluation of the method for improving the durability of the blocking die. For the macroscopic analysis, next to the typical tools used for macroscopic tests, the 3D scanning technique was applied, which is at present successfully used for the evaluation of the wear of tools removed from the process, mainly owing to their easy analysis and interpretation of results. In the analysis of the wear of the blocking die, over 50 tools were scanned, and the obtained scan images were compared with the CAD model of the tool. On this basis, a color map of deviations was created, from which it is possible to determine the areas of the biggest wear. Additionally, for a selected group of tools, every 100th forging was collected from the process in order to precisely track the wear mechanisms occurring on the tool during the operation. The studies conducted so far make it possible to divide the analyzed dies in operation I into two groups. The first group includes those that became worn during the process initiation—during the attempt at extrusion of the charge material, the forging was blocked in the tool without the possibility of continuing the production process. The other group of tools is constituted by the blocking dies that have been worn, and the destructive mechanisms occurring on them run according to a specific and repeatable pattern. In order to better present the wear mechanisms of the blocking dies, one tool was selected, which had produced the minimal number of items, i.e., 1, and one die represented the maximal tool wear, i.e., one that had made 2350 items. The tool that had produced one item, i.e., T1, was removed from the process as the forging became stuck in the blocking die. In the presented scan of the tool, we can clearly see the sticking of the material (probably of the forging) up to the thickness of 0.04 mm (Figure 4a). In the same scan image of the slug forging, we can clearly notice material loss up to a thickness of even 0.95 mm on the formed stem, where the whole stem surface has a wavy texture. On the tool’s cylindrical section, no wear traces were recorded.

The tool representing the second group of blocking dies is T2350, which made 2350 items (Figure 4b). While analyzing the particular sections of the tool, we can notice that, in the cylindrical part, the material loss is the smallest. Next, at the bottom of the cylindrical part, we can see a ring with a negative deformation value up to −0.25 mm. Shifting downwards, there is a ring with a positive deformation reaching 0.3 mm. The biggest material loss is in the section forming the stem, which even reaches −0.68 mm, on the measured half of the tool. The pattern of the wear occurring on blocking die T2350 was used to introduce a division into three zones, characterized by different wear mechanisms—Figure 4c shows the assumed division. In zone no. 1, plastic deformation and thermo-mechanical wear are dominant. Zone no. 2 represents material growth (resulting from the sticking of the forging material) as well as thermo-mechanical fatigue. Zone no. 3 is characterized by abrasive wear and plastic deformation. Figure 4d presents the results of the so-called reverse 3D scanning, consisting of scanning every 100th forging, consecutively collected from the production process. Such a compilation enables an analysis of the history of the tool’s operation based on the change in the forgings’ geometry. On the final forging, a big material growth occurs on the stem (over 0.7 mm), which, in this case, decides to remove the tool from further operation.

In addition to the macroscopic observations of the surface, an analysis was also made of the microstructure of the tools’ surface layer with respect to the presence of cracks and other defects as well as the precipitations of the alloying elements. The analysis included the blocking dies T1 and T2350, selected on the basis of the previous tests. After the forging of one item, we can observe pull-outs of the tool material as well as abrasive wear in the stem-forming area (zone no. 3 according to Figure 4). On the remaining part of the tool, slight traces of wear can be seen (Figure 5a). Another tool was removed from the process after producing 2350 forgings—we can see the formation of numerous wear mechanisms on it (Figure 5c). In area no. 1, there are significant plastic deformations in the form of axial grooves and a loss of shape in the section shaping the preform, visible by the edge of the examined forging. Both in area 1 and especially in area 2, we observe a strongly developed network of thermo-mechanical fatigue cracks (2.0 × 4.0 mm). The surface in area 3 (according to Figure 5d) is strongly plastically deformed. There are visible oxide impurities in the lubricant as well as numerous cracks in the tool material (0.5–1.0 mm), which are strongly branched and deformed. All this proves that the substrate material has been tempered, which was confirmed in the hardness tests.

The wear analysis performed on a light microscope confirmed numerous cracks in the tool material at the surface reaching as much as 0.8 mm into the blocking die T2350 (Figure 5c,d). The biggest cracks are in the area where the stem is formed (zone 3)—in some areas in this zone, the phenomenon of shearing of the formed material parallel to the direction of its flow was also observed during forging. This is also an effect of the strong adhesion of the forging material to the die surface. Additionally, there is no visible nitrided layer. Probably, after its removal, the tool’s degradation process proceeded very rapidly. In zone 2, there are also visible cracks, but they are much shallower, and their insides probably contain oxides, which were formed as a result of the presence of high temperatures, as well as deformations on the tool surface.

Additionally, an evaluation of the microstructure was made by means of a light microscope (Figure 6a), which revealed the thick sticking of the Nireva material in the stem-forming area (zone no. 3 according to Figure 4). During the forging process, the phenomenon of the shearing of the formed material parallel to the direction of its flow can be observed. This is a result of the predominance of the material that will adhere to the die surface over the internal cohesion of the formed material. The microstructure at the top of the tool, in area no. 1, points to the presence of a thin nitrided layer below 100 μm. There is no recorded wear in the remaining part of the tool. Next, on blocking die T1, an EDS analysis was completed in order to confirm the presence of the forging material on the tool (Figure 6b). The presence of Ni at the level of 30%, 15% Cr, and 2% Ti unequivocally points to the Nireva material.

For tools T1 and T2350, the hardness distributions were made by the Vickers method with a load of 0.1 in the indicated areas on the sample (Figure 7). The blue diagram presents the results for blocking die T1, whereas the orange one is made for tool T2350.

For zone 1 and tool T1, we can determine the thickness of the nitrided layer at the level of up to 0.1 mm. In turn, in the case of tool T2350, the nitrided layer was probably removed during operation. Additionally, the substrate material underwent tempering to a hardness below 400 HV0.1. For zone 2, the hardness of tool T1 is over 1300 HV0.1, and in the consecutive measurement points, the hardness is at the level of that of the native material, which is about 550 HV0.1. In the case of die T2350, the nitrided layer was probably sheared, as the first measurement point, falling to the depth of 0.02 mm from the tool surface, has the value of 800 HV0.1. A similar situation is presented in the case of zone 3 for this tool. In zone 3, for tool T1, the first two hardness values are 440 HV0.1. This is the hardness of the forging material, which became stuck in the area where the stem was formed.

### 3.2. Numerical Modeling

In order to perform a more thorough analysis and determine the parameters and physical quantities difficult to determine experimentally, numerical modeling was conducted. By means of a program for numeral simulations, Forge NxT 3.0, and based on the measurements and tests performed under the conditions of the real production process, a model was constructed to be used in the analysis of tool wear. The forging material from which motor truck engine valves are made is Nireva, and, since the material database of the Forge program does not contain this steel, a very similar grade was chosen, X5NiCrAlTi31-20/1.4958, for which the material data were assumed. The boundary conditions necessary for the numerical simulations include the strength properties, which, for the Nireva steel, were experimentally determined in a universal simulator of metallurgical processes, i.e., the Gleeble 3800 device (DSI Europe GmbH, Tatschenweg 1, D-74078 Heilbronn, Germany). In the deformation scope of 0–1, for the deformation rates of 0,1; 1; 10; and 40 [1/s] and at the four temperatures of 850; 950; 1050; and 1100 [°C], the yield stress dependence curves were determined in the function of the deformation of the Nireva material (Figure 8a). For the analysis of the first operation, an axisymmetrical model was made, which was used for the numerical simulations (Figure 8b).

The yield stresses in the function of deformation were determined experimentally for the Nireva material. By means of a non-linear estimation, owing to the specific curves on the Gleeble 3800 device, the Spittel equation was determined (1), which assumes the following form [48]:(1)$$\sigma_{f}=Ae^{m1T}T^{m9}\varepsilon^{m2}e^{\frac{m4}{\varepsilon }}(1+\varepsilon {)}^{m5T}e^{m7\varepsilon }{\dot{\varepsilon }}^{m3}{\dot{\varepsilon }}^{m8T}$$
where:ε—total deformation;$\dot{\varepsilon }$—deformation rate tensor;𝑇—temperature;A, m1, m9, m2, m4, m5, m7, m3, m8—model coefficients dependent on the material.

The Spittel equation determined the material data describing the mechanical properties of Nireva in the Forge program under the conditions of hot forging. The bodies deformable in the simulation, that is, those that enable the determination of stresses on the tool as a result of contact, e.g., with the forging, are the forging, the punch, the die, and the casing. In turn, the slide, the base, and the clamping ring are non-deformable bodies. Next, the heat exchange coefficients between the forging and the punch were assumed to be 8 kW/m^2^·K, between the forging and the die 2 kW/m^2^·K, and between the tools 1.5 kW/m^2^·K. As the tool material, steel X0CrMoV5-1/1.2344 from the Forge database was selected. The initial temperature of the forging was 1050 °C. For the die, the determined temperature was at the level of 380–390 °C, and these data were compared with the measurements made under the real process conditions. The entire deformation process took 0.1 s. The program specified a complete displacement of the punch, dividing the entire distance of several millimeters into a single computational step of 0.5 mm. The program divides the time steps in such a way as to count the entire deformation process in the stages of the punch movement every 0.5 mm. The simulation was created as axisymmetric with trio elements. The size of the elements is 1 mm, but in the zone of intense deformation, their size is 0.6 mm. The initial number of elements for the sample (input material) was 22,684 and changed during the calculations due to deformation and also due to the reconstruction of the mesh. Additionally, in order to obtain more accurate calculations, automatic remeshing for smaller elements was adopted, which was activated at an element deformation of 0.2. A blocking die was constructed (Figure 9a) with a drilled opening in the lower part of the tool for the thermocouple. Next, the production process was carried out with the thermocouple introduced into the tool, and the temperature results in the die were recorded (Figure 9b), which are analogical to those in the numerical simulation (Figure 9c). During the calculation of individual forging cycles (temperature changes), a more accurate simulation of one cycle is first calculated, and then the remaining cycles are calculated on its basis until the process temperature is established. This type of simulation is only for temperature calculations.

In order to select the proper friction coefficient in the numerical simulation program, a roller/preform was prepared and cut in half. On one half, horizontal lines at even distances of 5 mm were grooved by means of a laser (Figure 10a).

Both halves were welded together and then introduced into the production processes. After the extrusion of the stem in the first seat, clear flow lines were formed on the slug form, which were compiled with the numerical model. Based on the obtained results, a numerical model with analogical flow lines was prepared (Figure 10b,c), for which the determined friction coefficient was at the level of 0.3. For the preparation of the model of tools used in the simulation program, ready geometrical tool models were used. The data of the Maxipres 700 E press were assumed as follows: a crank length of L = 576 mm and double cranks of R = 127 mm, which were considered in the numerical modeling. By means of the numerical simulation program, the reduced stresses as well as the normal stresses on the die surface were determined (Figure 11). In the case of reduced stresses, the highest value is 1400 MPa and falls in the area that begins the forming of the stem. For normal stresses, the highest value equals 1200 MPa and also falls in the area where the narrowing of the tool begins. For both distributions, the highest stresses occur in the same area. We can presume that this area can be dominated by two destructive mechanisms, i.e., thermo-mechanical fatigue and plastic deformation. In order to confirm this, another model was constructed, where the die material behaves like an elastic–plastic body. Additionally, the contact time of the forging after the first deformation operation was determined (Figure 11).

For both distributions—of the plastic deformation and the forging’s contact with the tool—after the first forging operation, we can confirm that the area where the highest values were obtained overlaps with the maximal values of the reduced stresses and normal stresses. During the analysis of the wear mechanisms on the tool removed from the production process, we can expect that the area dominated by wear mechanisms, such as thermo-mechanical fatigue and plastic deformation, is the area in which the forming of the valve stem begins. The next distribution determined in the numerical simulation program is the distribution of the abrasive wear according to the Archard model as well as the friction path (Figure 12).

We can see that the high friction path from the FE simulation falls in the area of the biggest narrowing on the tool, where the stem is already shaped. The friction path in this area equals 60 mm and the abrasive wear is 0.15. The results presented above should be treated as the assumed initial state of the process (what should be obtained if the assumed conditions were ensured). Due to the low repeatability and stability of the industrial process, numerical models were constructed, which can show what can happen if the assumed conditions are different. To that end, five different variants were assumed, which, in the authors’ opinion, can significantly contribute to premature wear of the forging dies. The following variants were assumed: I. Nominal process; II. Increased friction from f = 0.4 to f = 0.6; III. Lowered charge temperature up to 950 °C; IV. Increased charge temperature up to 1150 °C; V. Increased tool temperature up to 300 °C. Figure 13 shows the global results of the numerical modeling in the above-listed variants.

Among the variants selected for further analysis, i.e., the normal stresses, the most advantageous seems to be the distribution for the nominal variant (at the level of 650 MPa), whereas high pressures occur for variant III, that is, for the temperature of the charge lowered to 950 °C, which, with a raised yield point for such a material, can really cause increased stresses (about 950 MPa) as well as a hindered material flow (Figure 14a). In turn, the tool temperature is, of course, most affected by the increase in the tool’s initial temperature from 200 °C to 300 °C. With this said, the “net” difference with respect to the initial value for variant V equals about 100 °C. In turn, among the remaining variants, for which the initial temperature was 200 °C, the biggest “net” difference is for variant II, where the temperature in the working area increased by over 150 °C, that is, by about 350 °C.

Increasing the input material temperature by 100 °C also raised the temperature of the tool by about 120 °C. Probably it had a positive effect in that it lowered the forging forces in this variant (Figure 14b).

The presented test results for both the nominal process and the processes that may occur in reality (multi-variant FEM simulations) as a result of an improperly followed technology, pointing to big changes in stresses and temperatures in the tool surface layer, have convinced the authors to consider the possibility of changing the charge material to an alternative one—as a cheaper and relatively easy-to-analyze method of improving the forging tool’s durability. So, a decision was made to apply other materials belonging to the group of tool steels dedicated to forging dies, with the simultaneous preservation of the previous nitrided layer at the level of 0.15 mm.

### 3.3. Selection of a Hot Operation Steel for the Blocking Die with the Nitrided Layer

The wide scope of hot operation steels available on the market has been narrowed down to three steel grades with the following commercial names: QRO90 Supreme, Unimax, and W360 (Voestalpine, 4020 Linz, Austria). The chemical compositions of those steels have been included in Table 1. The data come from the material specifications given by the steel producers.

The Unimax and W360 steels have a higher carbon content than the QRO90 Supreme steel, which gives them slightly worse ductility and thermal fatigue strength. Together with the increase in the manganese content, the steel’s impact strength improves. A very strong carbide-forming element is vanadium, which forms simple MC-type carbides, characterized by a high level of hardness and stability at high temperatures, increasing the effect of the secondary hardness. Molybdenum increases the effect of the secondary hardness and delays the processes taking place during tempering. Chromium favors the formation of complex M_23_C_6_-type carbides with lower stability at high temperatures, which coalesce at elevated temperatures. The indicated differences in the chemical composition can be significant in the aspect of tool wear.

Dies made of the three steel grades, after the standard thermal treatment., i.e., hardening and two-fold tempering as well as nitriding to a diffusion layer thickness of 0.15 mm, were introduced into the production process. The tools before the process were heated to 200 °C in a chamber furnace and then mounted in the seat of the press. The charge material had a temperature of about 1050 °C, and the blocking dies were lubricated and cooled with a lubricant based on graphite. Each test was performed on ten tools. A detailed analysis was conducted on the wear mechanisms examined on a representative tool from the given group. Figure 15 shows scan images of the blocking dies after the operation, where each of the analyzed tools had worked over about 2000 items, which is near the average value.

The average wear values for the applied tools are similar, where slightly better results were obtained for QRO90 Supreme. This can be caused by a better adjustment of the chemical composition to the conditions of the process. For each area on the tool, the material loss as well as the growth looked similar, where, in the cylindrical part, the biggest growth was recorded on Unimax, reaching over 0.11 mm. Next, a clear blue ring, proving a material loss in the analyzed area, the highest value (−0.33 mm) was obtained on the blocking die made of W360. This ring is formed as a result of the gravitational fall of the charge material into the tool. The last analyzed area of stem formation has clear grooves and furrows, where the biggest material loss falls on the tool made of Unimax and reaches −0.34 mm. Table 2 presents the global results referring to the performed tests.

The selected tools underwent an analysis with the use of a scanning electron microscope in order to evaluate the state of the surface. The analysis showed similar wear mechanisms for all the introduced tools. Figure 16 illustrates the consecutive blocking dies made of QRO90 Supreme, Unimax, and W360. In turn, the successive images (a), (b), and (c) are areas 1, 2, and 3, as shown in Figure 4.

Area 1 is mainly characterized by radial cracks running around the tool. Area 2 is dominated by a network of thermo-mechanical cracks with numerous spallings of the tool material. The visible dark spots are the remains of the graphite after the carried-out process. In the narrowing, i.e., area 3, there are visible traces of abrasive wear as well as the dark stick-ons of the lubricant and minor cracks. Additionally, for the tool made of Unimax, in areas 2 and 3, we can observe plastic deformations of the tool material. Tests with the use of a light microscope point to numerous and diversified wear mechanisms, especially in the tool’s surface layer. Starting from area 1, according to Figure 4, that is, the lower cylindrical part of the tool, there are visible microstructural changes characterized by intensive etching—this is connected with the presence of a nitrided layer (Figure 17a, Figure 18a, and Figure 19a). The light-colored area at the surface is the decarburization of the material. The numerous spallings and cracks in area 1 lead to the mechanical removal of the diffusive layer. The observed cracks are localized only in the area of the nitrided layer. Another area subjected to analysis, that is, no. 2, is the area on the tool in which there is a change in the diameter and that is, for all the applied hot operation steels, characterized by relatively uniformly distributed cracks, typical of thermal fatigue, parallel to the working surfaces (Figure 17b, Figure 18b, and Figure 19b).

In the cracks, there are products of oxidation and the remains of the lubricant. The cracks have a transcrystalline character. There is also an area of intensive etching, analogical to area 1. Additionally, for the blocking die made of Unimax for area 2 (Figure 18b), a narrow band parallel to the working surface was observed, which was etched more intensively. This is probably caused by the presence of tensile and shear stresses. Area 3, the area with the most intensive wear, underwent the strongest degradation, with numerous cracks and spallings (Figure 17c, Figure 18c, and Figure 19c). There is a visible decarburization at the surface, especially for Unimax and W360. Stick-ons of the forging material were also observed.

Microhardness tests were conducted for the selected areas, i.e., 1, 2, and, 3, as shown in Figure 4, for the blocking dies made from QRO90 Supreme, Unimax, and W360 and coated with a nitrided layer to the diffusive layer thickness of 0.2 mm (Figure 20).

The biggest differences can be seen in area 1. The QRO90 Supreme material, in the first point, still has the hardness of the nitrided layer. For the Unimax material, a clear drop of hardness was observed to have a value lower than the core material, which is caused by the disintegration of the nitrided layer as well as material decarburization. In turn, the lowered hardness of W360 in area 1 is caused by material tempering. For area 2, a similar trend was observed for each applied material, with a local drop in hardness of the nitrided layer caused by the cyclic heating of the tool material. The course of hardness in area 3 for QRO90 and Unimax is similar in character, its drop being probably caused by material decarburization, whereas the hardness of W360 in area 3 has a character that is analogical to that in areas 1 and 2. After the analysis of the results for the three applied materials, QRO90 Supreme was selected, with a nitrided layer thickness of 0.2 mm, as it has the highest average durability, and the number of extruded forgings in the die has the smallest scatter.

The applied QRO90 Supreme material, with a nitrided layer thickness of 0.2 mm, is characterized by better performance properties at elevated temperatures, causing better durability for the tools made from this material. The technological tests performed under industrial conditions demonstrated that the average die durability after the introduction of this durability-improving method (QRO90 Supreme) increased by 1000 items and equals 2200 forgings, compared to the data presented in the diagram in Figure 3b. The maximal value possible to obtain for the given blocking die also increased and equaled 2900 items. On this basis, it was decided that the best solution would be the use of the QRO90 Supreme material for the blocking dies.

## 4. Summary and Conclusions

In this section, the performed investigations have demonstrated that the process of manufacturing a valve forging is a production technology that is complex and difficult to analyze or improve. By itself, the first operation of the two-stage process, including co-extrusion, has already shown a lot of technological issues, which are of key importance for the durability of the dies used in this operation. The conducted tests have made it possible to draw the following conclusions.

−The analysis of the dies demonstrated that the average durability of the tools examined throughout a longer period of time is at the level of 1200 forgings, where, among the analyzed tools, we can distinguish two groups. The first is constituted by dies that have become damaged during the process initiation—during the extrusion of the charge material, the forging became stuck inside the tool, without the possibility of continuing the production process (usually, during the extrusion of the first forging). The other group of tools are dies whose operation time was within the scope of 1000–2000 forgings, and the destructive mechanisms occurring on them run according to a specific and repeatable pattern. A detailed analysis of the problem of tool wear during the process initiation has been discussed in [15]. In turn, in the case of the other group of dies, with significant durability, the macroscopic analysis showed that, for each die, we can distinguish a few similar characteristic areas/zones on the cross-section.−The key zone, as far as premature wear of the tool is concerned, is the area of the second radius of reduction outside the curvature (areas 2 and 3). In this zone, there are visible longitudinal scratches, which resemble propagating cracks. A probable cause of such a state is the fact of the preheated preform falling into the impression together with small particles of the scale (hard oxides of the forging’s metal). Another cause of the formation of scratches can also be the detachment of bigger, also hard, particles (nitrides) from the nitrided layer, as a result of, e.g., an improper thermo-chemical treatment—there are no data on the type and conditions of the nitriding process or the thermal treatment of the dies. The scratches and grooves formed in this way can cause increased surface roughness and thus also increased friction in this area. This, in turn, may favor the non-cyclic sticking of the forging material.−For a more thorough analysis, especially of the geometrical changes in the normal direction with respect to the CAD model, scanning was performed by means of a laser scanner. Based on the scanning results, we can observe that the changes taking place within the working area of all the tools are non-cyclic; that is, they are not repeated on the circumference of the axisymmetrical dies. This mostly concerns the material growth as a result of the sticking of the forging material. The material excess observed on the scan images, reaching as much as about 0.2 mm, most probably comes from the forging material.−The analysis performed based on the microstructural tests has demonstrated both numerous layers of stuck-on material with a thickness of 230–300 μm and a diffusion layer of nitrides with a thickness of about 100–200 μm, where, in different areas of the tool, the thickness of the layer is different. The nitride precipitates in some areas are more visible while being slightly covered with the layer of the stuck-on material, which hinders the etching of the microstructure.−An in-depth analysis of the die surfaces showed the presence of shallow scratches and abrasive grooves localized in the region of the die’s “neck”, i.e., where the cross-section of its opening becomes smaller—the key zone in the vicinity of area 3. In this zone, as a result of the contact of the forging material with the tool as well as the operation of the tensile stresses in the surface layer, we observed the phenomenon of longitudinal cracking in the direction along the die axis as well as the phenomenon of deepened cracks and abraded grooves and whole fragments of the die surface. In these areas, the surface of the die becomes black and mat (instead of smooth and shiny), and due to the increase in its roughness, the friction coefficient rapidly rises in contact with the formed material. These phenomena lead to the adhesion of the forging material to the tool surface, which, in consequence, leads to excessive narrowing of the die opening and improper shaping of the forging material.−The conducted investigations with the use of scanning microscopy have also demonstrated the presence of numerous grooves and scratches pointing to the phenomenon of abrasive wear, especially in the key zone of the die. Slightly lower, as demonstrated by the tests, we can also observe sticking of the forging material. The stuck-on material comes from forging, which was confirmed with a test of the chemical composition by means of an EDX-type X-ray analyzer integrated into the microscope. An effect of this adhesion is a change in the geometry of the tool’s cross-section through the narrowing of the die opening, which causes the low quality of the forging surface as well as the increased friction of the formed material with the formed layer, and, in critical moments, even blocking of the forging and the difficulty of its removal.−The performed numerical simulations of the analyzed forging process showed that, for the assumed conditions of the industrial process, the tools should not undergo wear or damage that fast. An important issue is the control of temperature, as, due to a small thermal capacity, resulting from the small “size” of the charge, as well as dies in the case of forging process instability, the conditions can entirely change. This can be seen especially in the case of the distribution of the temperature field on the forging during extrusion, where a small thermal capacity caused big differences in the forging temperature in the deformed section as well as in contact with the cooler tools. A similar situation refers to the tribological conditions, particularly the assurance of the optimal lubrication of the die, because, as shown by the simulation results, in the case of increased friction, the shaped forging material flows with less ease, and also the forging force of the process increases.−Additionally, due to the small repeatability and stability of the industrial process, numerical models were elaborated, and then computer simulations were carried out, which enabled a more thorough analysis, especially in the case (which can happen) when the assumed conditions are different. To that end, five different variants were assumed, which, in the authors’ opinion, can significantly contribute to the premature wear of the forging dies. In addition to the nominal process, the following variants were assumed: increased friction; lowered and increased charge temperature; and increased tool temperature.−The obtained test results as well as the in-depth analysis, especially for the nominal process, enforced the replacement of the tool material, due to the fact that the alternative is one of the cheapest and relatively easy-to-analyze methods of improving the durability of forging instrumentation. A decision was made to use and compare three materials belonging to the group of tool steels, dedicated to forging dies for hot operations. The selection was narrowed down to three steel grades with the commercial names QRO90 Supreme, Unimax, and W360, with the simultaneous preservation of the previous nitrided layer at the level of 0.15 mm.−The tests conducted under industrial conditions showed that the best results, compared to the other materials, were obtained for QRO90 Supreme, with a nitrided layer thickness of 0.2 mm. This is because, per 10 dies made of each material, the obtained average die durability was at the level of only 2200 forgings, which makes it possible to state that the tool made of this material is characterized by better performance properties at elevated temperatures with respect to the dies made of the other two materials. An additional important point was that of the economical aspect, as, among the tested materials, the price of QRO90 Supreme steel was 5–20% lower than that of the other two. On this basis, it was decided that the best solution would be the use of QRO90 Supreme as the material for the dies.

## Figures and Tables

**Figure 1 materials-17-00346-f001:**
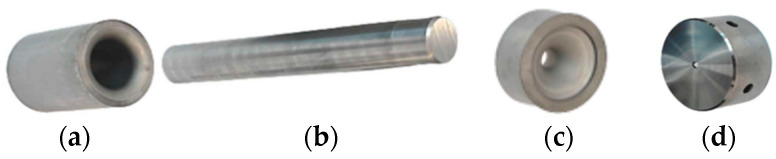
View of (**a**) the die in operation I, (**b**) the punch in operation I, (**c**) the die in operation II, and (**d**) the punch in operation II.

**Figure 2 materials-17-00346-f002:**
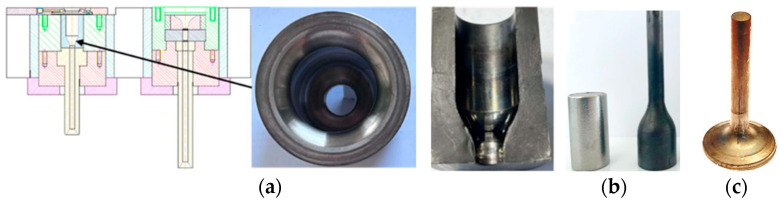
View of (**a**) the lower bench of the press with extrusion and forging, (**b**) the blocking die (top view), the cross-section of the blocking die, (**c**) the initial material, and the forgings after the first and second operation.

**Figure 3 materials-17-00346-f003:**
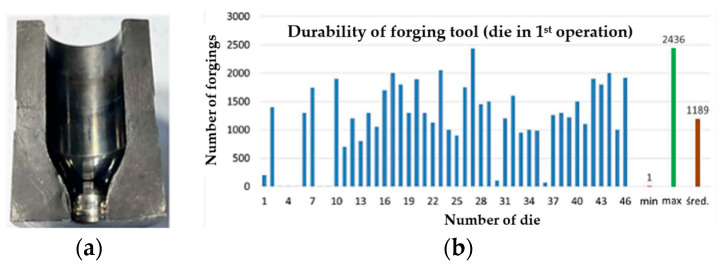
View of (**a**) a cross-section of the die in operation I and (**b**) the diagram of the blocking die’s wear [47].

**Figure 4 materials-17-00346-f004:**
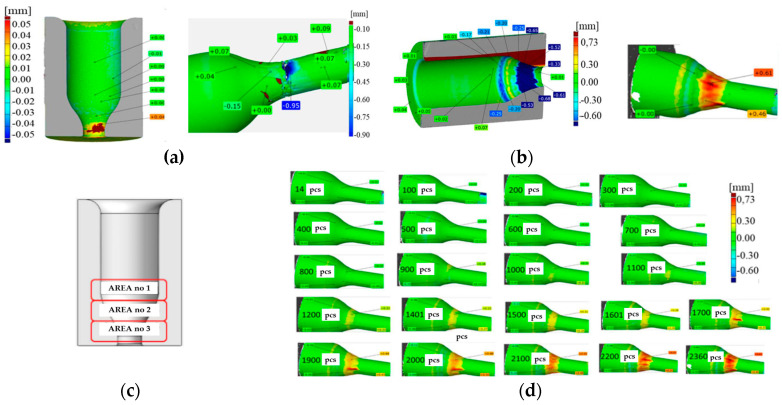
View of (**a**) the result of the comparison of the 3D scan of the T1 tool’s half with the CAD model, (**b**) the scan image of the last forging from the T2350 tool, (**c**) the tool with marked characteristic zones, and (**d**) 3D scans of periodically sampled forgings for subsequent stages of tool wear [47].

**Figure 5 materials-17-00346-f005:**
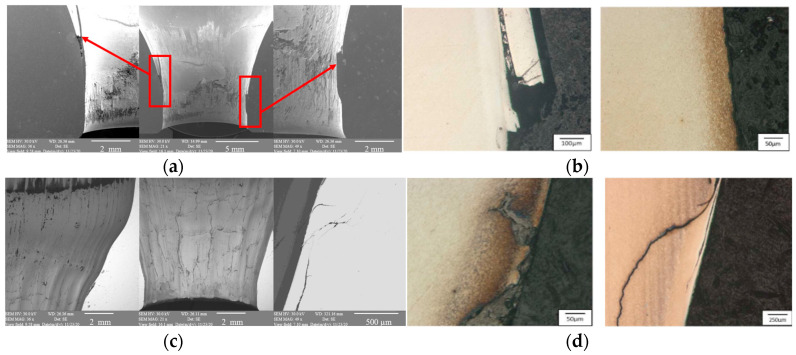
View of (**a**) the microstructure of the blocking die with the Nireva material stuck in the stem-forming area (zone no. 3), (**b**) the microstructure of the blocking die in the upper part of the tool (zone no. 1), (**c**) the SEM image of the selected working areas of T235, (**d**) the microstructure of the blocking die using light microscopy and in its etched state.

**Figure 6 materials-17-00346-f006:**
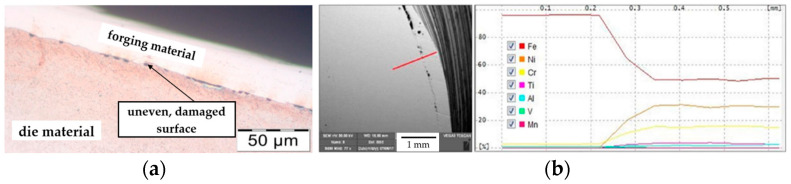
View of (**a**) a photograph of the die from areas 2–3 in the cross-section together with a visible sick-on of the forging material and (**b**) the linear element distribution EDS obtained on the cross-section along the line marked in the SEM area.

**Figure 7 materials-17-00346-f007:**
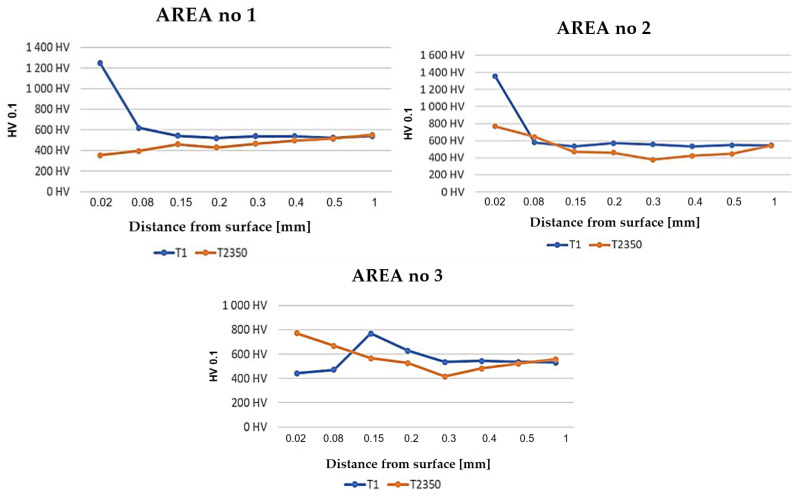
Hardness distributions for the selected areas in the representative dies.

**Figure 8 materials-17-00346-f008:**
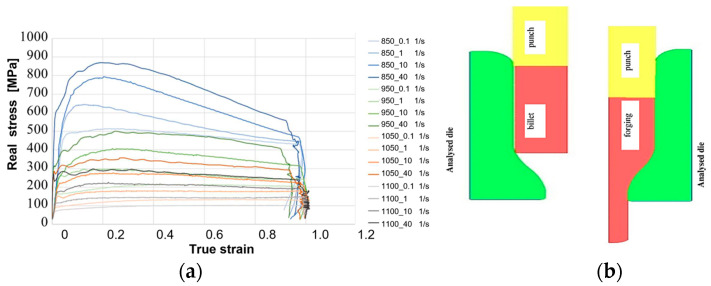
View of (**a**) the yield stresses in the function of the deformation determined experimentally for the Nireva material and (**b**) the FE model before and after one forging operation.

**Figure 9 materials-17-00346-f009:**
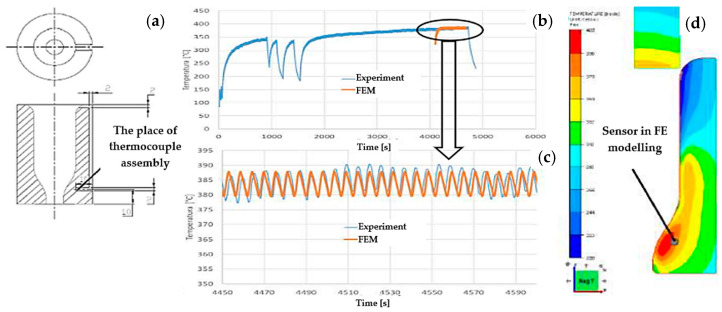
View of (**a**) the draft of the blocking die with a drilled opening for the thermocouple, (**b**) the real measurements of the temperature on the die, (**c**) the temperature distribution after 70 cycles with the marked area of the temperature measurement, and (**d**) the FEM results of the temperature field distribution.

**Figure 10 materials-17-00346-f010:**
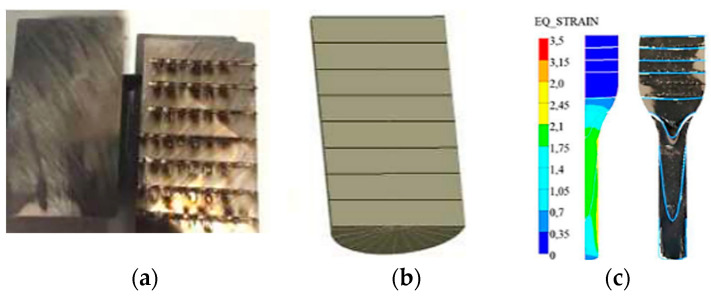
View of (**a**) the cut preform and real sample, (**b**) the sample prepared in the simulation program, and (**c**) FEM—the plastic deformation distribution with the plotted flow lines on the forging, the flow lines from the simulation (blue) plotted on the photograph of half of the forging with the drilled grooves after 1 deformation operation.

**Figure 11 materials-17-00346-f011:**
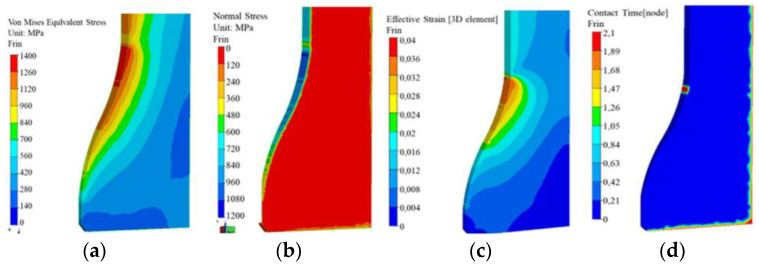
Distribution of the stresses: (**a**) reduced, (**b**) normal on the blocking die, (**c**) the plastic deformations on the die, (**d**) the contact time after one operation of forging a valve.

**Figure 12 materials-17-00346-f012:**
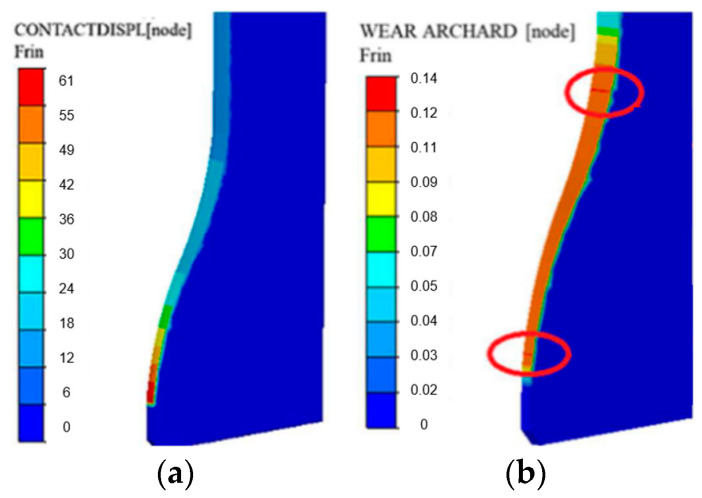
View of (**a**) the friction path and (**b**) the abrasive wear distribution according to the Archard model (the highest values are marked with red circles).

**Figure 13 materials-17-00346-f013:**
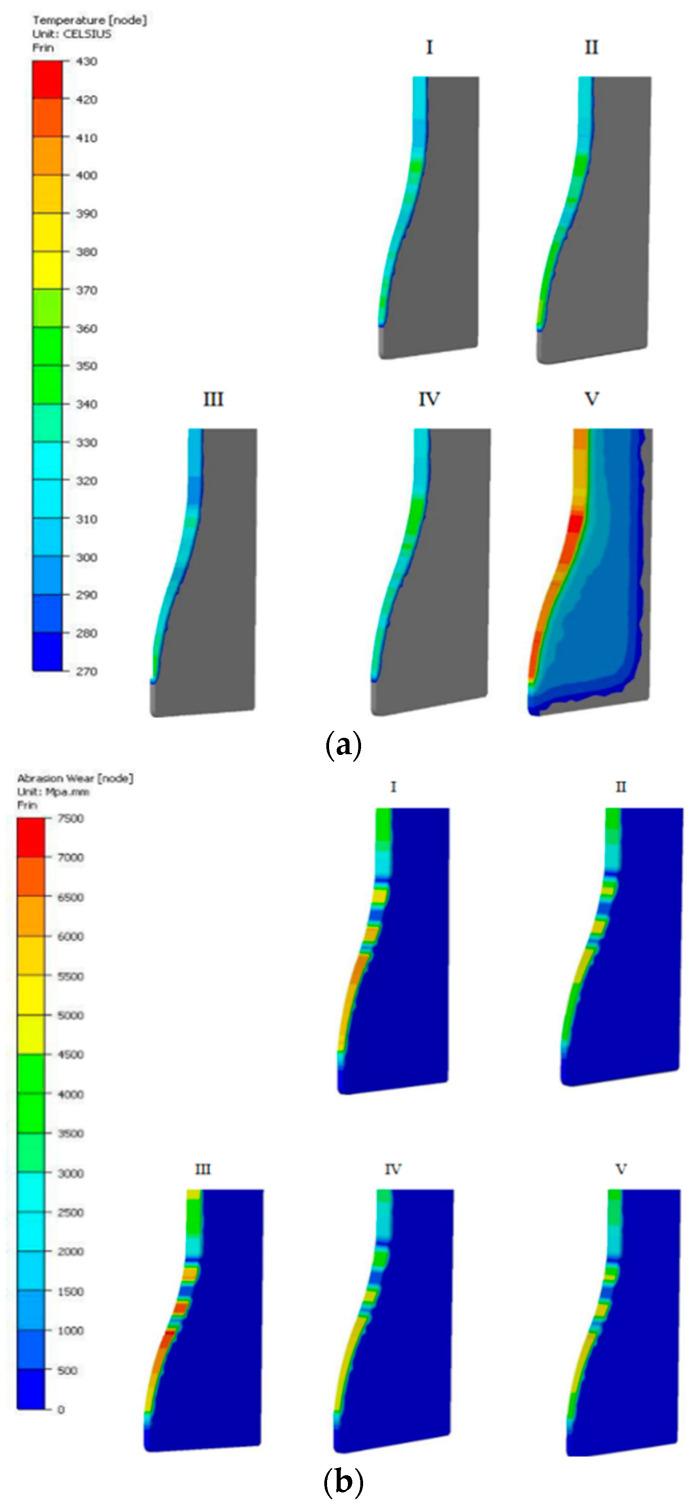
Global results of numerical modeling for the 5 selected variants: (**a**) the temperature field distributions and (**b**) the wear distributions according to the Archard model.

**Figure 14 materials-17-00346-f014:**
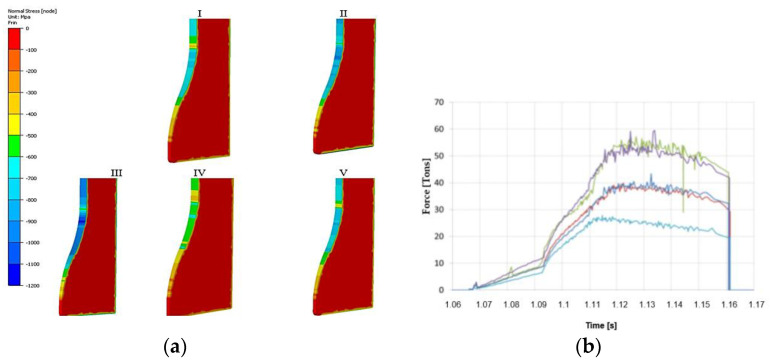
The view of (**a**) the normal stress distributions and (**b**) the forging force courses as a function of time for the particular variants.

**Figure 15 materials-17-00346-f015:**
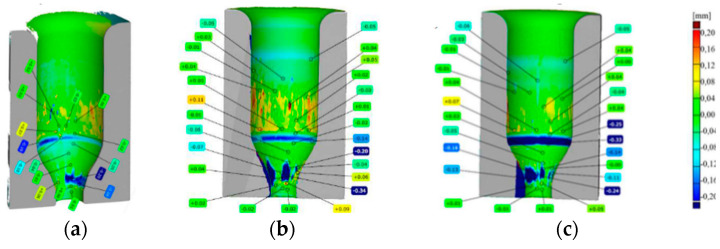
A comparison of the scans with maps of geometrical deviations for the dies made of (**a**) QRO90 Supreme, (**b**) Unimax, and (**c**) W360 [47].

**Figure 16 materials-17-00346-f016:**
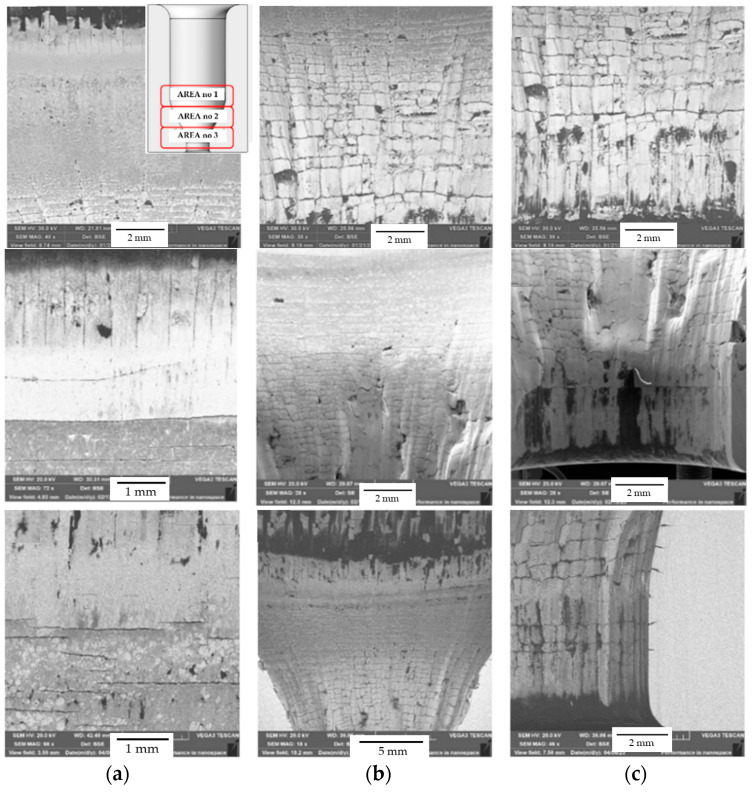
The results of the SEM images in the sub-surface area of a cross-section made on the blocking die after the operation for the successively selected materials: QRO90 Supreme; Unimax; W360: (**a**) area 1, (**b**) area 2, (**c**) area 3, SEM [47].

**Figure 17 materials-17-00346-f017:**
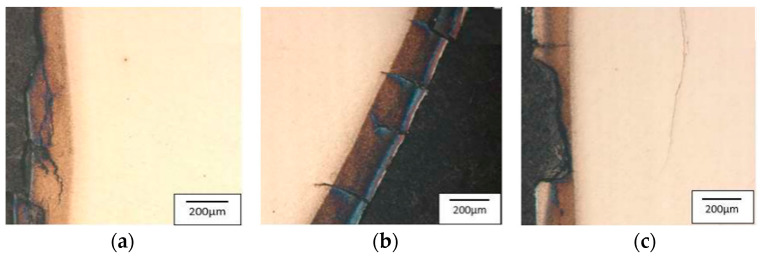
Photographs of the blocking die surface after the operation for QRO90: (**a**) area 1, (**b**) area 2, (**c**) area 3. Light microscopy, etched state.

**Figure 18 materials-17-00346-f018:**
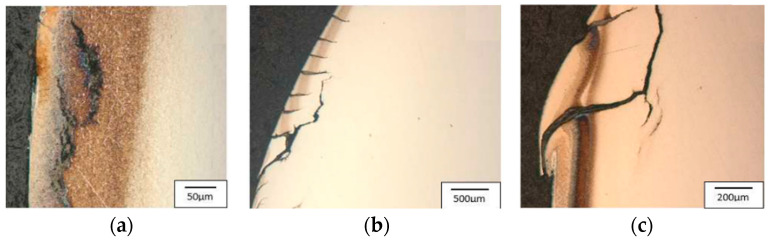
Photographs of the blocking die surface after the operation for Unimax: (**a**) area 1, (**b**) area 2, (**c**) area 3. Light microscopy, etched state.

**Figure 19 materials-17-00346-f019:**
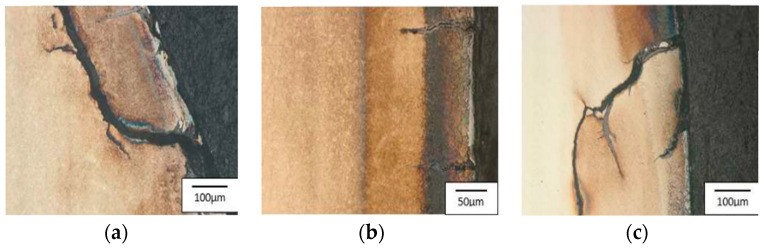
Photographs of the blocking die surface after the operation for W360: (**a**) area 1, (**b**) area 2, (**c**) area 3. Light microscopy, etched state [47].

**Figure 20 materials-17-00346-f020:**
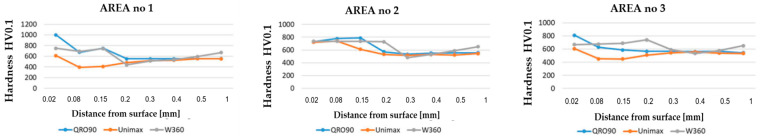
The hardness distribution of the blocking dies made from the three different materials for areas 1, 2, and 3.

**Table 1 materials-17-00346-t001:** Chemical composition of hot-work tool steels used for preliminary dies.

Element	C [%]	Si [%]	Mn [%]	Cr [%]	Mo [%]	V [%]
QRO90 Supreme	0.38	0.30	0.80	2.60	2.30	0.90
Unimax	0.50	0.20	0.50	5.00	2.30	0.50
W360	0.50	0.25	0.50	4.50	3.00	0.60

**Table 2 materials-17-00346-t002:** List of tools used in the research.

Materials	Hardness [HRC]	Nitrided Layers [mm]	Durability of Dies [pcs]
QRO90 Supreme	53	0.15–0.16	2100 (min: 1960; max: 2218)
Unimax	53	0.14–0.16	1870 (min: 1550; max: 2480)
W360	56	0.15–0.17	1830 (min: 1685; max: 2430)

## Data Availability

Data are contained within the article.

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
