# Peer review of "Possibilities of Increasing the Durability of Dies Used in the Extrusion Process of Valve Forgings from Chrome-Nickel Steel by Using Alternative Materials from Hot-Work Tool Steels"

_materials, 2024, doi:10.3390/ma17020346_

Round 1

Reviewer 1 Report

Comments and Suggestions for Authors

First of all, I would like to thank you very much for choosing our journal for your article. It is a very successful and meticulously prepared article. If you answer the questions I have asked, I would like to read the article again.

- Could you provide more details on how the wear of the dies and punches was quantitatively measured during the real production process?

- Could you elaborate on how the standard thermal treatment and gas nitriding process contribute to the durability of the tools?

- How did the microstructural changes observed through light microscopy and SEM correlate with the wear observed in the tools?

- How accurate were the 3D scanning and comparisons with CAD models in assessing tool wear?

- How did the performance of these alternative materials compare to the original WLV material in terms of durability?

- What are the potential challenges in implementing these changes in a standard production setup?

- Are there any other aspects of the forging process that could benefit from similar detailed analysis?

- In your introduction, please mentioned these two important studies about combustion technology

Thippeshnaik G, Prakash SB, Suresh AB, Chandrashekarappa MPG, Samuel OD, Der O, Ercetin A. Experimental Investigation of Compression Ignition Engine Combustion, Performance, and Emission Characteristics of Ternary Blends with Higher Alcohols (1-Heptanol and n-Octanol). Energies. 2023; 16(18):6582. https://doi.org/10.3390/en16186582

Technological evolution of internal combustion engine vehicle: A patent data analysis

https://doi.org/10.1016/j.apenergy.2021.118003

- Can you explain the methodology used to select the tools for analysis, particularly why the tools with the minimal and maximal wear values were chosen?

- How accurate and reliable is the 3D scanning technique in detecting and measuring wear compared to traditional methods?

- What were the main findings from the microstructural analysis of the tool's surface layer, and how do these findings correlate with the observed wear patterns?

- Can you provide more details on the observed phenomenon of material shearing and its impact on the tool wear?

- Were there any variations in the thickness or composition of the nitrided layer across different tools, and if so, how did these variations affect wear?

- How does the presence of elements like Ni, Cr, and Ti in the forging material influence the wear and durability of the tools?

- Are there any other aspects of tool wear or material properties that you plan to investigate in future studies?

- Can you explain the significance of the differences in nitrided layer thickness between tool T1 and tool T2350, and how this impacts their respective wear patterns?

- What conclusions can be drawn from the varying hardness levels in different zones of the tools, especially in relation to the wear mechanisms observed?

- Could you elaborate on the choice of the X5NiCrAlTi31-20/1.4958 steel for the simulation in place of Nireva, and how closely these materials compare?

- Can you discuss the findings related to the friction path and abrasive wear distribution, particularly in the area of the biggest narrowing on the tool?

- Among the variants studied, which one most closely represents the actual conditions observed in the production process, and why?

- Are there any specific changes to the forging process or tool materials that you suggest based on the results of your study?

Author Response

Dear Reviewer,

Thank you for your reviews and valuable comments. Detailed answers to individual questions/comments have been provided and included in a separate file (attachment).   regards,

Reviewer 2 Report

Comments and Suggestions for Authors

A good introduction was noted; however the scientific novelty and authors contribution was noted as limited

In the methodology section please provide details of lubrication used and the set up for lubrication

“…material mass with the precision of +/–1-2%...” OK this sentence but what about geometrical tolerance ?

“..which the minimal and maximal values..” OK, but what about the average that are more relevant situation ? I mean about considering the averaging condition which were not considered

“..adhesion forces of the forging material..” indeed is about adhesion but not the forces will adhere !- please revise this formulation; the material will adhere

“on it (Fig. 59).” Please check the entire manuscript for different typos, as here you don’t have figure number 59; also the degree symbol should be well adjusted

Figure 8 scare bar is almost invisible please adjust it for better clarity ; other wise please use the same measure as now you have 50 um against 250 um

Figure 9 requires better quality as the font sizes are poor quality

Figure 9 requires standard deviation

Boundary conditions and loading details of numerical model are required to be presented in this manuscript

The speed used in Figure 10 is equal to one used in real industrial condition ?

“the Spittel equation was determined” a ref is required for ref 1

Please revise most /all images for quality and font sizes as for some of them the font sizes and legends are practically invisible for difficult to read

Comments on the Quality of English Language

some english improvement are required 

Author Response

(The authors gave the same response as above.)

Reviewer 3 Report

Comments and Suggestions for Authors

This is quite an extensive passage discussing the selection of hot operation steel for blocking dies in the context of valve forging. The study seems to focus on the analysis of different steel grades (QRO90 Supreme, Unimax, and W360) and their performance in terms of durability and wear under specific operational conditions.

The study considered three steel grades with different compositions: QRO90 Supreme, Unimax, and W360. QRO90 Supreme showed slightly better results in terms of ductility and thermal fatigue strength. The tools underwent standard thermal treatment, including hardening, tempering, and nitriding. Tests were performed under specific conditions, such as heating the tools and using lubrication based on graphite. Wear mechanisms were examined, and the average wear values for the tools were similar, with slightly better results for QRO90 Supreme. The study includes detailed scans and SEM images of blocking dies made of the three steel grades after operation. Microstructural tests revealed layers of stuck-on material and a diffusion layer of nitrides. The analysis also highlighted areas prone to cracking, scratches, and abrasive wear. Microhardness tests were conducted for different areas of the blocking dies, showing variations in hardness related to the nitrided layer and material characteristics. The authors have concluded that the QRO90 Supreme was selected as the preferred material for blocking dies due to its better performance properties at elevated temperatures, resulting in increased durability compared to the other materials.

It seems like a comprehensive study with detailed analyses and testing to arrive at a practical and economical solution for improving the durability of forging instrumentation. The manuscript is well written and shows interesting results and discussions. Furthermore, it shows potential originality. However, the novelty of the research is not clearly emphasized. Below is a list of suggestive revisions that might help improve the manuscript.

1. What is the main question addressed by the research?

The main question addressed by the research seems to be the selection of an optimal hot operation steel for the blocking die in the production of valve forgings. The study aims to identify the most suitable steel grade among QRO90 Supreme, Unimax, and W360, considering factors such as chemical composition, wear mechanisms, and performance properties at elevated temperatures. The goal is to enhance the durability and efficiency of the tools used in the forging process.

2. Do you consider the topic original or relevant in the field? Does it address a specific gap in the field?

The topic appears to be relevant in the field of manufacturing and materials engineering, specifically in the context of hot forging processes for valve production. The research addresses the selection of an appropriate steel grade for blocking dies, considering various factors like chemical composition, wear mechanisms, and performance at elevated temperatures. While I don't have information on the current state of the field, the study seems to provide valuable insights into optimizing tool materials for hot forging applications, which could be particularly important for industries involved in valve manufacturing. Whether it addresses a specific gap would depend on the existing literature in the field, which I don't have access to beyond my last training cut-off in January 2022.

3. What does it add to the subject area compared with other published material?

The research adds to the subject area by systematically evaluating and comparing three different steel grades (QRO90 Supreme, Unimax, and W360) for blocking dies in hot forging processes. The study considers various factors, such as chemical composition, wear mechanisms, thermal fatigue strength, and impact strength. By conducting detailed analyses, including microstructural tests, microhardness tests, and scanning electron microscope examinations, the research provides a comprehensive understanding of the wear patterns and material performance during the hot forging process.

This level of detail and the focus on practical applications, including the industrial testing of the selected materials, contribute valuable insights for practitioners and researchers in the field of hot forging. The study's emphasis on both material properties and performance under industrial conditions enhances its practical relevance, potentially offering a more holistic perspective compared to other published materials that might focus on specific aspects of the hot forging process.

4. What specific improvements should the authors consider regarding the methodology? What further controls should be considered?

The methodology of the research seems robust, but there are a few aspects that the authors might consider for improvement:

4.1-Transparency in Nitriding Process Details: The study mentions the nitriding process, but specific details such as the type and conditions of the nitriding process are not provided. Including more information about this process could enhance the reproducibility of the study.

4.2-Controlled Environmental Factors: The study acknowledges the importance of temperature control and lubrication in the forging process. To further strengthen the methodology, the authors could consider additional controls or experiments to investigate the effects of variations in these factors systematically.

4.3-Extended Comparative Analysis: While the study effectively compares the three selected steel grades, it might be beneficial to extend the comparative analysis by including more materials or conducting experiments with variations in the processing parameters. This could provide a more comprehensive understanding of the material selection process.

4.4-Long-Term Durability Studies: The research mainly focuses on the average durability of the tools, but including a more extended study on the long-term durability and performance degradation over a larger number of forgings could offer valuable insights.

4.5-Detailed Cost Analysis: The study briefly mentions the economical aspect, indicating that the price of QRO90 Supreme was lower. A more detailed cost analysis, considering not only the initial material cost but also factors like tool life, maintenance, and overall economic impact, could strengthen the study's practical implications.

4.6-“Experimental section” needs include the ISO/ASTM standard procedure citation for metallography, mechanical testing (tensile) and so forth. For example, the standard geometrical size of the tensile specimens needs to be clearly mentioned and/or provided as a figure. Moreover, it needs to include detailed explanation of metallography (such as the grinding paper types, forces) as well as technicality of the electron microscopy (such as the working distance, kV, and so forth). Also, figure 1 seems to be not clear. I would suggest making the image parts in the figure larger in visual scale than the text font size!

These suggestions aim to enhance the robustness and applicability of the research methodology.

5. Are the conclusions consistent with the evidence and arguments presented and do they address the main question posed?

The conclusions drawn in the research appear to be consistent with the evidence and arguments presented throughout the study. The authors have systematically analyzed the wear mechanisms, conducted tests with different steel grades, and thoroughly examined the microstructural changes in the tools. The selection of QRO90 Supreme with a nitrided layer of 0.2 mm is well-supported by the data, indicating better performance properties and increased durability under industrial conditions.

The conclusions directly address the main question posed in the research—identifying the most suitable steel grade for blocking dies in the valve forging process. The decision to use QRO90 Supreme is justified based on its average durability, extruded forgings, and cost-effectiveness compared to the other materials tested.

Overall, the conclusions align with the objectives of the study and the evidence presented, providing a clear resolution to the main question addressed.

6. Are the references appropriate?

Almost.

7. Please include any additional comments in the tables and figures.

Below are some suggestive revisions that may improve some of the result and discussion sections:

7.1- I would suggest adding a brief section in the introduction and explaining the numerical modeling with proper explanation and referencing.

7.2- In general, the number of figures (25!?) is too many for a scientific journal paper. They need to be reduced to about 10 figures in total. For instance, it seems that figures 4, are 5, are repeating each other and could be easily combined to a summarizing version of all with a few parts. Similarly, figures 6, are 8, are repeating each other and could be easily combined to a summarizing version of all with a few parts. Likewise, figures 13, 13, 15, and 16 are repeating each other and could be easily combined to a summarizing version of all with a few parts.  figures 18, 19, and 20 can be combined. Likewise, figures 21, 22, and 23 can be combined.

7.3- There are two figures 13!? It seems to be typo and the second figure 13 is the figure 14 because the next figure is figure 15.

7.4- Figure 5 is too busy and needs to be reduced in number

7.5- Figures 9 and 24 need to be presented vertically and revised in appearance in a more appropriate format for a scientific journal paper. Example is the axes are not clear. The font sizes are small. The markers are similar for different conditions.

7.6- Figure 10 (a) is very busy. I would suggest presenting the entire figure vertically. The part (a), the curve needs to be clearer in terms of their legends, such as their font size, etc.

7.7- Figure 15 is also very busy and needs to be reduced in number of parts.

Author Response

(The authors gave the same response as above.)

Reviewer 4 Report

Comments and Suggestions for Authors

A study of the durability of dies used in the exhaust valve extrusion process was conducted using various materials, including chrome-nickel steel and alternative materials such as hot work tool steels.

It is a comprehensive work that includes experimentation, micrographic analysis, finite element simulations, and SEM micrographs.

To enhance the quality of the article, I suggest making some minor changes.

In Figure 1, the appendices a and b appear to be upside down. Specifically, the die is labeled as figure a and the punch is labeled as figure b.

In section 3, I consider that as it is explained, not only the results obtained from each analysis are being commented, but also a discussion is being made, so I would name it "Results and Discussion".

There are figures, such as 4, 11 (b), and 17, where I recommend modifying the legends for 11b and enlarging the figures. The labeling of the color labels cannot be read, and I believe that this would improve the quality.

Regarding subsection 3.2. When discussing numerical modelling, it is important to provide details on both the mesh and solution module parameters. Specifically, the mesh size, type and size of elements, and number of elements should be clearly explained. Additionally, the parameters of the solution module used in the software, such as the time step or simulation time, should be described. This level of detail is particularly important in severe plastic deformation processes, where careful consideration of these parameters is necessary to ensure reliable results.

With these revisions, I believe the article is ready for publication.

Comments on the Quality of English Language

The language of the article should be improved.

Author Response

(The authors gave the same response as above.)

Round 2

Reviewer 2 Report

Comments and Suggestions for Authors

.

Comments on the Quality of English Language

.

Author Response

Dear Reviewer,

Thank You once again for your review.

regards,